# Methods to Improve Accuracy of Electronic Component Positioning in Thermoformed Electronics

**DOI:** 10.3390/mi14122248

**Published:** 2023-12-16

**Authors:** Behnam Madadnia, Jan Vanfleteren, Frederick Bossuyt

**Affiliations:** Centre for Microsystems Technology, Imec and Ghent University, Technology Park 126, B-9052 Ghent, Belgium; jan.vanfleteren@ugent.be

**Keywords:** thermoforming, 3D-shaped electronics, thermoformed electronics, in-mold electronics, component positioning, structural electronics

## Abstract

Three new methods for accurate electronic component positioning for thermoformed electronics are presented in this paper. To maintain the mechanical and electrical properties of printed-ink tracks, prevent deformation and stretching during thermoforming, and ensure reproducibility, the component positioning principle for all three proposed methods is based on keeping the temperature of some regions in the thermoplastic substrate less than the glass transition temperature of the thermoplastic carrier, to keep those regions resistant to plastic deformation. We have verified the accuracy of the different approaches by implementing these methods in a semi-sphere mold for positioning seven LEDs and one printed capacitive touch sensor. We compared the result of our fabrication processes with the typical fabrication process of in-mold electronics (direct printing on a thermoplastic foil and followed by a thermoforming step) and noticed that the sample produced by the typical process had tracks that were randomly stretched, tracks were not in a straight path after thermoforming and they were not electrically conductive. Furthermore, the final 3D position of the components was not reproducible sample by sample. However, with our proposed fabrication methods, the tracks and pads do not deform or expand during thermoforming and are electrically conductive after. Moreover, the round shape of the touch sensor remains the same as in the 2D design. Based on the results of the experiments, it appears that the proposed methods are capable of positioning electronic components with high precision in thermoformed electronics.

## 1. Introduction

Over several years, 3D-shaped electronics have become more popular, finding multiple applications in manufacturing, home appliances, and the car industry. A number of methods, including in-mold electronic technology (IME) [1,2,3,4,5,6,7], stretchable mold interconnect (SMI) [8,9,10,11,12,13], and 3D-MID [14], can be used to produce curve-shaped electronics. IME is a technique that combines conventional plastic molding with printed conductive inks that are flexible or even elastic on plastic surfaces. For SMI technology, the 3D form is made possible by using thermoforming and flexible metal conductors. Commonly utilized procedures like copper plating and laser cutting are used in the 3D-MID method. Vanfleteren, J. et al., examined and compared three techniques; their outcomes are discussed in [15]. During thermoforming in SMI and IME technology, the spatial position of the components is difficult to control. Due to the complicated deformation of 2D substrates caused by thermoforming, implementing a repeatable process is extremely challenging. The electrical components in thermoformable electronics can be positioned using the “Degrees of freedom” methodology, according to a patent application by Vanfleteren, J. Using this method, the meandering will only direct the component in one possible spatial position [16]. Gong, Y. researched resistance changes depending on printed circuit deformation during thermoforming. To explain the association between deformation and the electrical resistance of the circuit, a regression model is developed. The deformation of the printed circuit is measured experimentally and statistically. Following this, a theoretical analysis is conducted based on numerical findings and a regression model to determine the electrical resistance distribution of the circuit following the thermoforming process. Both experimentally and numerically, the impact of geometrical parameters is also studied [17]. Ting, J et al., developed software with Microsoft that allows users to map 2D data into 3D space, design electrical circuits and structural models, and interact with the software [18]. The paper utilized mathematical mapping to convert 2D-generated circuits into 3D forms. It is possible to interactively view the updated position of the component in 3D by changing its position in 2D. Research in this area only examined the problem mathematically, disregarding the physics of thermoforming. A new 3D fabrication technique, developed by Choi, J., is adaptable, conformable, and stretchable. This technique is based on predistorted patterns and thermoforming. Using this technique, thermoforming is used to create custom-designed 3DE. Because the thermoforming method makes it possible to completely replicate both the overall shape and the surface texture of the 3D mold, the manufactured 3DE has great 3D conformability. A thermoplastic elastomer and a conductive electrode with a liquid metal base contribute to the device’s high thermoformability and stretchability during operation [19]. A study by Schirmer, J. evaluated the mechanical stress that is imposed on electronic assemblies during thermoforming and injection molding for conformable electronics. It has been shown using microscopic images that cracks may form near components after thermoforming [20]. Steven I. Rich presented a shrink-based paradigm to apply thin-film electronics to non-developable surfaces [21]. Steven I. Rich has reviewed all recent possible fabrication approaches for fabricating electronics on curved surfaces, e.g., substrate-mounted methods, aided patterning, fully 3D techniques [22]. Beltrão, M. reviewed the state of the art of IME technology, resorting to the scientific research work performed and its main outputs, compiling and analyzing the main knowledge and achievements on the process to date [6]. Goument, C. represented a solution to replace polycarbonate (PC) with poly(lactic acid) (PLA), a biosourced and biodegradable polymer to reduce the environmental impact of IME manufacturing. The thermal and mechanical properties of the PLA materials are investigated to optimize the IME process parameters and to take into account the glass transition temperature Tg around 55–60 °C of PLA [23]. Lall, P. investigated a significant design challenge, as the over-deformation of printed circuits can lead to a poor performance and reliability issues. This study has introduced new design strategies and methods to optimize the performance of the IME process. This includes the use of electrically conductive adhesives to attach components in their undeformed state and later subject them to thermoforming. This approach helps to maintain the integrity of the printed circuit and improve the performance of the interconnects [24]. 

It is difficult to industrialize thermoformed electronics since the spatial repeatability of electrical components positioning is one of the main barriers. To position components correctly in these thermoformed parts, it is necessary to determine where to place them in the 2D design so that after thermoforming, they end up at the correct position in the 3D-formed sample. According to our knowledge, there has not been much research conducted on electronic-component positioning in thermoformed and 3D-shaped electronics. In this article, we discuss three different novel methods for positioning the electronic components in thermoformed electronics. In the first approach, to maintain the mechanical and electrical properties of the printed conductive ink tracks, prevent deformation and stretching during thermoforming, and ensure reproducibility, we have used a heat-mask in the thermoforming machine to maintain the temperature of some regions lower than the glass transition temperature of the base thermoplastic substrate. During thermoforming, these parts of the substrate do not deform as long as they remain below their glass transition temperature. In the second approach, we laminated another laser-structured substrate to the base thermoplastic substrate having a higher glass transition temperature. As a result, during thermoforming this part of the substrate remains under its glass transition temperature and does not stretch. In the third approach, we print a reflective material on top of the printed tracks. During thermoforming, these parts of the substrate can maintain their mechanical properties as long as they remain below their glass transition temperature.

## 2. Methodology of Using a Metal Heat Mask

The principle of this method is to place a heat mask between the heat source and substrate. To accomplish this, we modified our thermoforming machine as depicted in Figure 1. The non-heated and thus non-stretchable regions of the substrate (indicated as “low temperature area” in Figure 1) are bendable and can accommodate the mold shape during forming; the exposed region of the substrate (indicated as “high temperature area” in Figure 1) will reach the glass transition temperature (glass transition temperature of the polycarbonate is around 147 °C) and can be stretched to accommodate to a non-developable mold shape. In order to achieve this selectivity, we have used a laser structured metallic mask plate (blocking window) to block heat (IR radiation) in specific areas and allow the heat to expose the substrate in other parts. In the set up depicted in Figure 1, there are two openings in the mask that can allow radiation to go through and expose the substrate to this radiation. 

As a proof of concept, we have conducted experiments to demonstrate how this mask can affect the thermoforming behavior of the thermoplastic substrate. Figure 2 represents a top view of the designed mask to block the IR radiation in the thermoforming machine. We have chosen a 3 mm aluminum material for the heat mask to have a durable mask, capable of withstanding the temperatures of the heating element during the thermoforming cycle.

Laser cutting of the aluminum plate was performed by an external company. The resulting heat mask is shown in Figure 3A. The thickness of the aluminum plate is 3 mm. In this experiment, we have used polycarbonate Lexan 250 µm (Lexan^®^ 9030, Vink, Belgium) as the substrate to be thermoformed. In our thermoforming set up, there is no gap between the metal mask and substrate. The distance between the IR heater and the mask is 10 cm. The exposure time of the substrate is 20 s. Figure 3B shows the polycarbonate substrate after it has been exposed to IR radiation through the mechanical mask. As can be seen, there are two distinct areas: exposed areas, where there is no aluminum part to block the radiation, and non-exposed areas, where there are aluminum parts to block the radiation. In the exposed area, the temperature will reach a higher temperature than the polycarbonate glass transition temperature. The reached temperature depends on the exposure time (e.g., for 50 s exposure time, the polycarbonate’s temperature will be ~160 °C). This region can deform then to accommodate the 3D shape of the mold. On the other hand, in the region where the polycarbonate is not exposed, the polycarbonate temperature will be lower than its glass transition temperature (lower than 40 °C). Therefore, the polycarbonate remains non-stretchable during the thermoforming process. Furthermore, based on the measurements, the strip width in the aluminum mask is 1 cm (Figure 3A), and the corresponding strip width on the polycarbonate after exposure is 1.4 cm. It is evident that the “unheated” or shielded PC areas seem to be larger than the mask. This is because the IR source consists of several heating elements, distributed over a large area, due to the masking effect of the aluminum plate (distance plate–substrate, thickness of the plate), there is no direct line-of-sight between a number of heating elements and certain expectedly exposed PC areas, close to the heating mask. This experiment shows that using an aluminum mask is an effective solution to selectively heat the polycarbonate. 

Based on [25], the non-stretchable regions in the substrate, created by heat shielding using an aluminum heat mask, can now be used to make electronic component positioning reproducible. 

To demonstrate this, we have realized different heat masks and printed electronic circuits as depicted in Figure 4.

We have used two different strip widths in the masks (i.e., $D_{1},$ $D_{2}$). In Figure 4, it can be seen that when a larger strip in the mask is used, some non-uniformity is present in the thermoformed sample, but when a smaller strip is used, there is no non-uniformity (e.g., in case A the thermoformed shape does not follow the shape of the mold, in case B it does). This method cannot provide a sharp transition between fully exposed and completely non-exposed areas. There is a gradual spatial change in absorbed heat, extending over several mm. The designs in Figure 4A,B seek to explore the limits of the spatial frequency or width of a non-exposed line which can be achieved. We have proven that a width of 3 mm of the metal heat mask still results in shielding the substrate from exposure to heating. As a result, we have used the optimized strip width ($D_{2}$ = 3 mm) for the rest of the experiment. 

For realizing the thermoformable substrates, we have printed conductive ink (DuPont ME614—Chimie Tech Services, Antony, France) on polycarbonate (Lexan 250 µm) by using a screen-printing technique (EKRA screen printing machine, ASYS, Benzstraße, Germany), followed by a curing step of 20 min in an air convection oven at 90 °C (Figure 5A). Conductive adhesive is applied on the contact pads (LOCTITE CE 3103WLV) (Figure 5B). Electronic components are mounted on the substrate and the assembled circuits are cured in the oven at 120 °C for 30 min (Figure 6C). Finally, the substrate is thermoformed with a semi-sphere mold (Figure 5D). We have used DuPont ME614 as the conductive ink, which should have the best thermoformability of DuPont’s inks (Table 1) [2]. The printed pattern is shown in Figure 6A. In the screen printer, we have used specific parameters to achieve high-quality printed tracks (Table 2). Figure 7B represents the printed and cured sample. To be able to investigate the deformation of pads, we did not mount the components on the substrate. Figure 7C represents the final thermoformed sample. Figure 7D represents a close-up of the printed tracks as a representation of the printing quality. We have checked the conductivity of the tracks after thermoforming and they are conductive. 

We have performed two different experiments by using two molds with different elongations. In the first experiment, we used a semi-sphere mold with 60% elongation (results in Figure 8). In the second experiment, we used a 16% elongation mold (results shown in Figure 9). Figure 7 shows the concept of the elongation in thermoforming. The following formula represents the elongation:$$\mathrm{Elongation}=\frac{\left|∆A\right|}{{Area\;}_{2D\;surface}}\;=\frac{\left({|Area\;}_{3D\;surface}-{Area\;}_{2D\;surface}|\right)}{{Area\;}_{2D\;surface}}\;=\frac{\left(|A_{2}-A_{1}|\right)}{A_{1}}=\frac{\left(12,628.96-7853.98\right)}{7853.98}\;=\;0.6$$

$A_{2}$ = area of deformed semi-sphere after thermoforming (including tracks);

$A_{1}$ = area of deformed semi-sphere before thermoforming, in flat state;

Elongation should be interpreted here as a relative increase in surface area.

In the first experiment, we have used a 60% elongation mold to thermoform the fabricated 2D stack. For the first set of samples, we have selectively irradiated the substrate by using the heat mask with the narrow (3 mm-wide) strips (Figure 4B), but for the second set of samples we have illuminated the whole region of the 2D substrate (no heat mask used) and we compared the results in Figure 8 and Table 3. As it is obvious from the comparison in Figure 8, when there is no heat mask used, the tracks are randomly stretched and they are not even in a direct path (i.e., straight lines are thermoformed to non-straight lines). There is also a big deformation of the shape of the sensor, in such a way that it is deformed differently in different directions. The pads of the components show a small amount of expansion (Figure 8). Furthermore, when no heat mask is used, there is no electrical connection between the powering pads and the LED’s tracks, because micro cracks in the ink appear which are caused by excessive stretching during thermoforming (Figure 10). For samples where a heat mask was used during the heating phase of the thermoforming, there is no deformation and expansion in the tracks and in the pads. Furthermore, the shape of the sensor is identical to the 2D design. The electrical conductivity of the tracks before and after thermoforming for both methods is depicted in Table 5. The average resistance before thermoforming is 5.67 Ω and after thermoforming 6.36 Ω. As can be observed, the tracks remain conductive for the method using a heat mask. Regarding accuracy of component position, in Table 3 we show that the dimensional properties of the polycarbonate in the heat-shielded areas remain the same as before thermoforming. As a result, the length of those non-exposed parts remains unchanged and can thus be used for accurate component positioning.

**Figure 8 micromachines-14-02248-f008:**
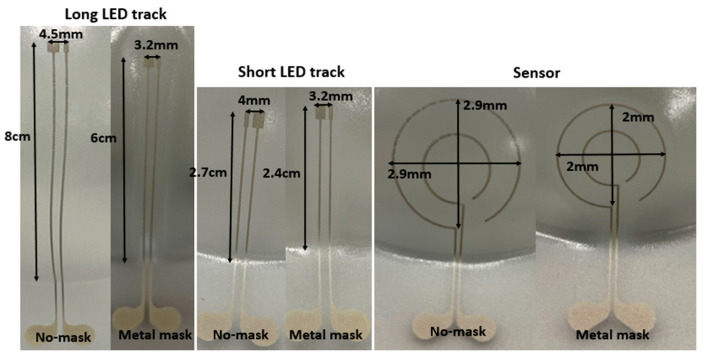
Measurement results of the thermoformed sample using a 60% elongation mold. When “No-mask” is mentioned, it means we have not used any mask to restrict the heat over the substrate in the thermoforming machine and “Metal mask” means we have used a specific mask, which is depicted in Figure 4B, to restrict the exposure of the substrate.

**Figure 9 micromachines-14-02248-f009:**
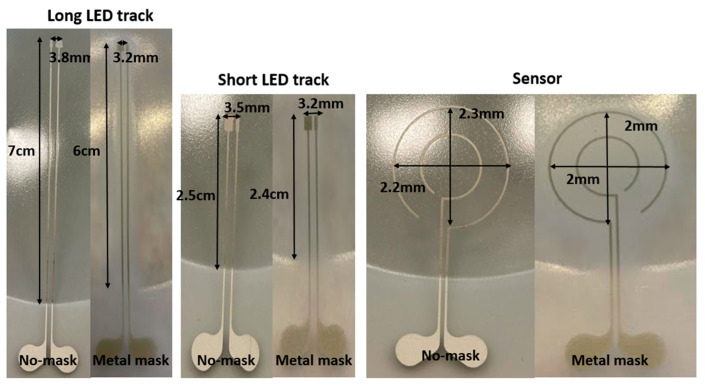
Measurement results of the thermoformed sample using 16% elongation mold. “No-mask” means we have not used any mask in the thermoforming machine to restrict the heat over the substrate and “Metal mask” means we have used a specific mask, which is depicted in Figure 10, to restrict the exposure of the heat to the substrate.

**Figure 10 micromachines-14-02248-f010:**
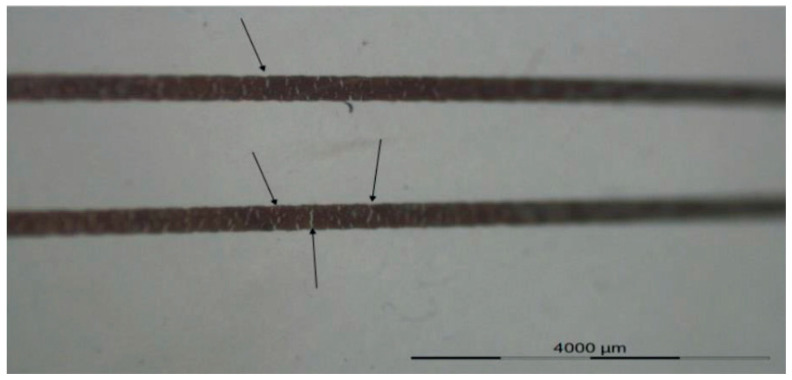
Micro crack formation as the result of the stretch of the ink during thermoforming by using a 60% mold elongation.

**Table 3 micromachines-14-02248-t003:** Measurement result on the thermoformed sample with 60% elongation.

	No-Mask	Metal Mask	2D Length before Thermoforming
Long LED track	7 cm	6 cm	6 cm
Short LED track	2.5 cm	2.4 cm	2.4 cm
Sensor diameter	2.2 cm	2 cm	2 cm
Long track pad	3.8 mm	3.2 mm	3.2 mm
Short track pad	3.5 mm	3.2 mm	3.2 mm

The results of the second experiment using a 16% elongation mold, are depicted in Figure 9 and Table 4. Based on the measurements shown in Figure 9 even with only the 16% elongation mold, we observe a difference between the lengths of the tracks in the samples in which we have used the heat mask, compared to those in which we have not used the mask. All tracks were still conductive after thermoforming for all sets of samples (with and without using the heat mask). We have used an automatic range digital multimeter (Coczow) to measure the resistance. The resistance measurements are listed in Table 5. The tracks’ length changed and deformation were observed in samples that were thermoformed without a heat mask in the forming machine. The length of the tracks and sensor shape remained unchanged in the samples that were thermoformed with a mask in the forming machine. 

## 3. Methodology of Using a Structured Foil Having a Higher Tg

In the previous section, the typical fabrication process for in-mold electronics is introduced. In this section, we aim to modify this fabrication process of direct printing thermoformable inks on thermoplastic substrates and thermoforming them (typical in-mold electronics process, as explained in previous section and Figure 5) and make component positioning more controllable using an innovative fabrication process (Figure 11). The process uses a substrate material with a higher glass transition temperature than that of the thermoformable polymer to remain mechanically stable during the thermoforming process. PET is chosen as the material with a higher glass transition temperature because it can retain its good mechanical properties at temperatures up to 175 °C. In our process, 250 μm PET ES301300 (Goodfellow Cambridge Ltd., Huntingdon, UK) will fulfill this role. In the first step of our process, the conductor ink is screen printed on the PET substrate. After printing, the printed sample is put in an oven to be cured at 90 °C for 20 min (Figure 11A and Figure 12A). The parameters that we have used for printing the conductive ink are described in Table 1. Then, the outline of the non-stretchable structure is realized by laser cutting (Figure 11A and Figure 12B). In the next step, the fabricated samples are laminated with TPU (Bemis—100 μm- Bemis Co-Lab UK, Aylesbury, UK) and polycarbonate (Lexan—250 μm Vink BV, Heist-op-den-Berg, Belgium) by using a vacuum lamination machine (Maschinenfabrik Lauffer GmbH & Co. KG, Horb am Neckar, Germany) with a specific pressure and temperature (Figure 11B and Figure 12C). In the next step, the components are mounted on the substrate by using conductive adhesive (LOCTITE CE 3103WLV, Henkel AG & Co. KGaA, Düsseldorf, Germany) (Figure 11D), but in this experiment the intention was not to mount the component to be able to investigate the contact pads’ deformation after thermoforming (Figure 11C and Figure 12C). Finally, the fabricated sample is thermoformed in the thermoforming machine by using a semi-sphere mold (Figure 11E). Figure 13 shows the final thermoformed sample.

In this experiment, we have used the mold with 60% elongation (same as in the first methodology). The measurement results are shown in Figure 14 and Table 6. By contrast, the tracks and pads do not deform or expand when we use the proposed fabrication method with the PET support. In addition, the sensor shape remains the same as it was in the 2D design. Table 6 demonstrates that by using our proposed approach all the dimensions of the tracks, pads, and sensors are equal to the 2D design, which proves that our approach is a repeatable process for the fabrication of thermoformed electronics. In our proposed method, the conductive tracks are still conductive for obvious reasons (there is no stretching in the tracks, there is just bending to accommodate the curved shape). In the new approach, the change in the average resistance before and after thermoforming is very small (i.e., 0.09 Ω), as shown in Table 7.

## 4. Methodology of Using a Printed Heat Mask

Typically, in the thermoforming process, the entire substrate is irradiated with light (e.g., with IR or other wavelengths) to be heated up to about the glass transition temperature to make the substrate non-resistant to deformation. Here, we have evaluated printing a reflective material on the substrate to selectively heat the substrate, as IR heat will be reflected. The difference in absorption causes the buildup of a differential temperature profile in the thermoplastic material along a surface coordinate of the substrate. Therefore, the regions with the printed reflective material can maintain a lower temperature than the surrounding area without this printed layer. At the time of the deformation process, the thermoplastic material at non-printed areas may have a relatively high temperature (i.e., above the glass transition temperature) to provide a relatively low resistance to the deformation process while the thermoplastic material at the printed parts has a relatively low temperature, (i.e., below the glass transition temperature) and provides a relatively high resistance to the deformation process. Figure 15 depicts how we can selectively heat the substrate.

To evaluate the effectiveness of this approach, we have printed a reflective material on a substrate and irradiated this substrate in the forming tool. The reflective material is a non-conductive mirror ink (Mirror Ink M3 N, Tiefschwarz, Pröll GmbH, Weißenburg in Bayern, Germany), which is applied to polycarbonate by screen-printing with a mesh mask. A metal mesh SD 90/40 screen (wire diameter 40 µm, mesh opening 90 µm, open mesh area 48%, 77 wires/cm) has been used for the screen printing. For better reflection properties, the printing needed to be performed two times. The reason why we printed two layers was the amount of pinholes still left in the layer after a single printing cycle and hence also the light which was going through these pinholes (Figure 16). Figure 17A,B represent the printed area on polycarbonate as the reflective region.

Figure 17 represents the measurement setup, including two thermocouples taped to the reflective area (Sensor B) and the pure polycarbonate area without a reflective layer (Sensor A). We have used the TC-08 temperature reader (Pico Technology, Saint Neots, UK) and its software to monitor the temperature during the heating process. According to the temperature profile, Sensor A will measure a much higher temperature than Sensor B, so by using this approach, we can maintain the region with the reflective layer in the rigid phase and resistant to deformation while allowing other areas without the reflective layer to reach the glass transition temperature and become non-resistant to deformation as shown in Figure 17C,D. 

The temperature of the two regions (i.e., under the reflective layer and polycarbonate) is measured (Figure 18). We heated the substrate for 45 s and then switched off the heating elements and let the substrate cool down. From Figure 19, we can see a 90 °C difference in the temperature of these two parts. As a result, we can ensure that in the region with the reflective layer, the material will not stretch during thermoforming, the material is bendable at the most; on the other hand, in the regions without reflective material, the material can reach its glass transition temperature to be thermoformed and accommodate the deformation. Figure 20 represents a schematic of the thermoforming process when there is a reflective layer printed on the thermoformable substrate. Based on [25], the non-stretchable regions in the substrate, created by reflective material, can now be used for component positioning. 

Based on these proof-of-principle experiments, we fabricated a number of samples to evaluate the component-positioning accuracy using this method. The fabrication starts with printing the conductive ink (ME614, CHIMIE TECH SERVICES, Antony, France) on polycarbonate (Lexan 250 µm) and then curing the samples for 20 min in the oven at 90 °C (Figure 21A and Figure 22A,B). The reflective non-conductive layer is printed on the substrate and is cured in the oven for 20 min at 90 °C (Figure 21B and Figure 22C,D). Then, a conductive adhesive is applied to the pads (LOCTITE CE 3103WLV) to mount the electronic components followed by a curing step at 120 °C for 30 min (Figure 21C,D). Finally, the substrate is thermoformed (Figure 21E).

We have used the same molds as in the previous methodologies (with 16% and 60% elongation mold). We compared samples without and with a reflective printed layer. The results of these experiments are shown in Figure 23 and Table 8. 

In the samples with reflective layers, there is a very small amount of expansion in the tracks and in the pads. The shape of the sensor is identical to the 2D design and there is no deformation. We have checked the electrical conductivity of the tracks which proved to be conducting. Table 10 represents the measurement data of the resistance of the tracks before and after thermoforming. The average resistance by using a 60% elongation mold is 4.86 Ω after thermoforming, compared to 4.09 Ω before thermoforming. As depicted in Table 8, we can observe that even by using the reflective material there is an expansion on conductive tracks. Thus, for a higher amount of elongation, we need to use a better reflective material to keep the temperature of the polycarbonate even lower than 70 °C.

In the second experiment, a 16% elongation mold was used and the results are described in Figure 24 and Table 9. Based on Figure 24, even with small amount of elongation there is difference between the track length in the samples where we have used a reflective layer compared to those where we have not used this layer. Table 10 represents the measured resistance of the tracks. The average resistance after thermoforming is 4.28 Ω, compared to 4.09 Ω before thermoforming. A length change and deformation of the printed tracks was observed in the sample where there was no reflective layer. In the sample with the reflective layer, the length of the tracks is equal to the 2D design. As a result, using this reflective layer a good positioning accuracy can be secured.

## 5. Discussion: Comparison between the Three Methods

### 5.1. Methodology of Using a Metal Heat Mask

This approach keeps the conductive track length the same as before forming. In addition, we are able to fabricate special patterns (e.g., sensors) and electronic component contact pads without deformation, which will help us to achieve reproducible positioning and fabrication of the 3D-shaped electronics by using a thermoforming process. In this approach, the 2D design complexity is very low. The designer just needs to measure the surface length of the mold (distance between an origin point and the component position on the 3D surface) and design the tracks on the substrate in straight lines and a mask in such a way that after applying this mask to the thermoforming machine, there will be a non-stretchable region in the 2D substrate (the same length as the measured length on the 3D surface). We could achieve a high accuracy (error < 1%) for positioning the electronic components. Regarding the electrical resistance of the tracks, there is maximum of 12% change when there is 60% elongation. There is no clear separation between exposed and non-exposed areas on the substrate; instead, a transition area of several mm form fully exposed to non-exposed areas is observed (Figure 3B). This means that high resolution patterns with a high spatial frequency (3 cm or higher) of exposed and non-exposed areas will not be possible with this method. The design complexity is restricted by the fact that the metal heat mask consists of a single mechanical piece.

### 5.2. Methodology of Using a Printed Heat Mask

This approach keeps the track length the same as it was before forming for a small degree of mold elongation (e.g., ≤16%). In addition, we are able to fabricate special patterns (sensors) and pads without any deformation. This method’s 2D design complexity is significantly low. It is only required to design the tracks on the substrate in straight lines and the reflective layer mask in such a way that, after the reflective material is printed on the substrate, there will be a reflective material region of the same length as the measured length on the 3D surface. We could achieve a low accuracy (error < 16%) for the positioning, but there is a limitation in the elongation and this approach is valid for an elongation smaller than 16%. Regarding the electrical resistance of the tracks, there is maximum of 18% changes when there is 60% elongation. Compared to the “metal heat mask method”, this methodology allows a higher spatial frequency of exposed and non-exposed areas (determined by the heat-mask printing resolution) and also allows for more complex designs, as multiple separate heat-mask areas can be printed on the same substrate. 

### 5.3. Methodology of Using a Structured Foil Having a Higher Tg

In this part, we applied the same concept of a non-stretchable structure for ink-based 3D-shaped electronics. We compared our approach with the typical fabrication process of in-mold electronics. The results show significant improvements from a positioning perspective compared to the typical processing approach. The 2D design is not too complex in this method. To construct the tracks on the substrate in straight lines and use the same length as the measured length on the 3D surface, the designer simply needs to measure the surface length of the mold (distance between an origin point and the component position on the 3D surface). The accuracy of the method is correlated with the non-stretchable part’s width. We could achieve a high accuracy (error < 1%) for positioning the electronic components. Regarding the electrical resistance of the tracks, there is a maximum of 18% change when there is a 60% elongation. A high spatial frequency of non-exposed areas can be achieved (comparable to the second method and determined by the printing resolution of the conductor ink). Similar to the second method, more complex designs are possible, as several pieces of structured foil can be integrated in the same thermoformable substrate. In summary, we can introduce the following table (Table 11) as the comparison between these three methods:

## 6. Conclusions

In this paper, we have provided three different novel methods for accurate component positioning in thermoformed electronics, based on the principle of having regions that do not deform, as the glass transition temperature of the polymers present in these areas is not reached during the thermoforming process. In this way, more accurate electrical component positioning can be achieved. In the first methodology, using a metal mask leading to selective heating of the substrate, there are no changes in the conductor track lengths and a maximum of 12% electrical resistance change is observed. In the second approach, where a laser-structured foil having a higher glass transition temperature than the surrounding thermoplastic material is used below the conductors, there are no changes in track lengths and a maximum of 2% electrical resistance change is observed. For the third approach, using a heat-reflecting printed layer in the regions of the conductors, there is a maximum of 16% change in track length and a maximum electrical resistance change of 18%. Furthermore, we are able to obtain undistorted special patterns (e.g., sensors) and electronic component contact pads after deformation, which will help to provide reproducible positioning and fabrication of thermoformed electronics.

## Figures and Tables

**Figure 1 micromachines-14-02248-f001:**
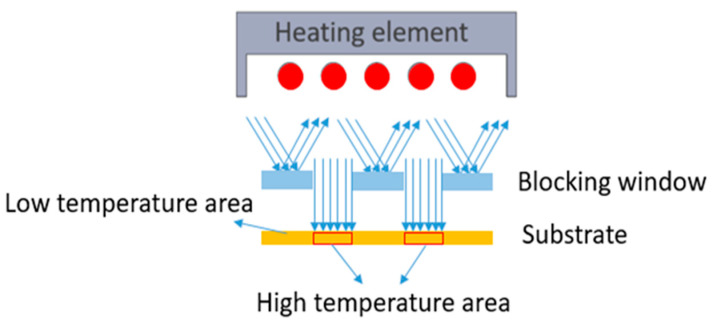
Using a blocking window to restrict the heat in the thermoforming machine.

**Figure 2 micromachines-14-02248-f002:**
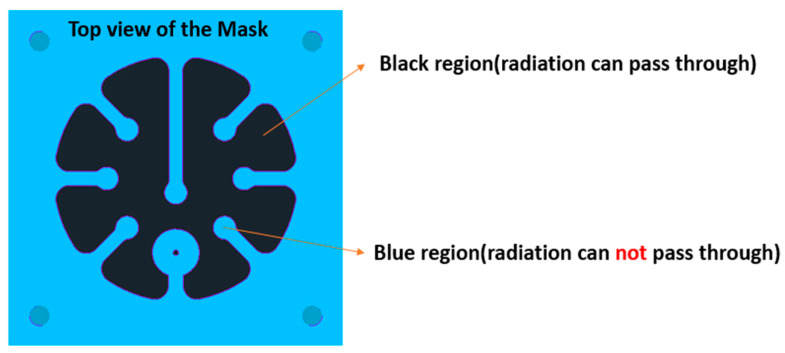
The blue region represents the area that IR radiation cannot penetrate, and the black area shows the areas exposed to IR radiation.

**Figure 3 micromachines-14-02248-f003:**
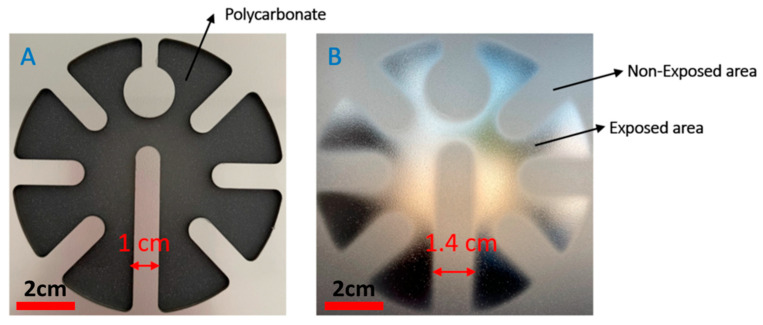
(**A**) The heat-blocking mask made in aluminum, (**B**) polycarbonate which has been exposed to IR radiation.

**Figure 4 micromachines-14-02248-f004:**
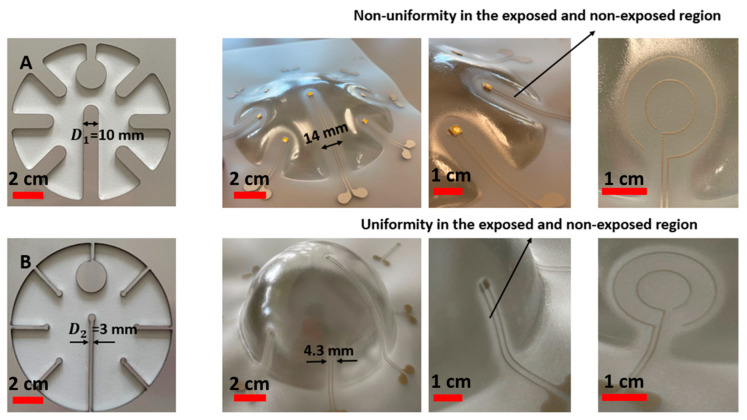
(**A**) Use of a 10 mm-wide strip in the metal mask and the resulting thermoformed sample, (**B**) use of a 3 mm-wide strip in the metal mask and the resulting thermoformed sample.

**Figure 5 micromachines-14-02248-f005:**
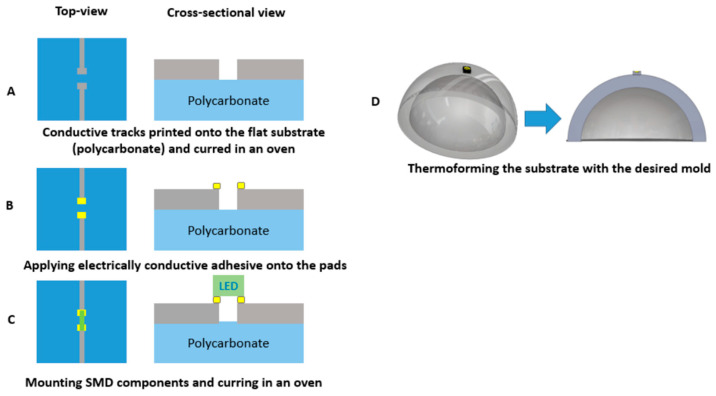
Fabrication process: (**A**) Printing conductive traces onto the flat substrate using ME614 and putting in the oven (90 °C) for 20 min. (**B**) Applying electrically conductive adhesive (LOCTITE CE 3103WLV)) onto the pads. (**C**) Mounting SMD components in putting in oven (120 °C) for 30 min. (**D**) Thermoforming the substrate with the desired mold (cross-sectional view).

**Figure 6 micromachines-14-02248-f006:**
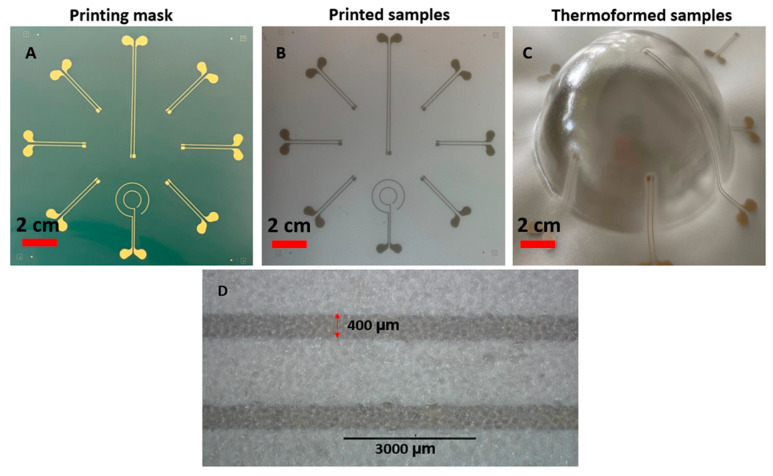
(**A**) Screen mask. (**B**) Printing result on the polycarbonate substrate using ME614 as the conductive ink. (**C**) Thermoforming result by using a semi-sphere mold with 60% elongation. (**D**) detailed view of 2 printed tracks (darker grey) after thermoforming.

**Figure 7 micromachines-14-02248-f007:**
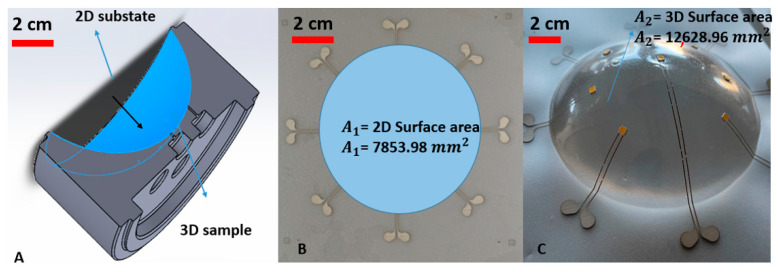
(**A**) A cross-sectional view of the 2D substrate and 3D sample surface area. (**B**) The 2D surface region in the substrate that is heated and formed. (**C**) The 3D surface area after thermoforming.

**Figure 11 micromachines-14-02248-f011:**
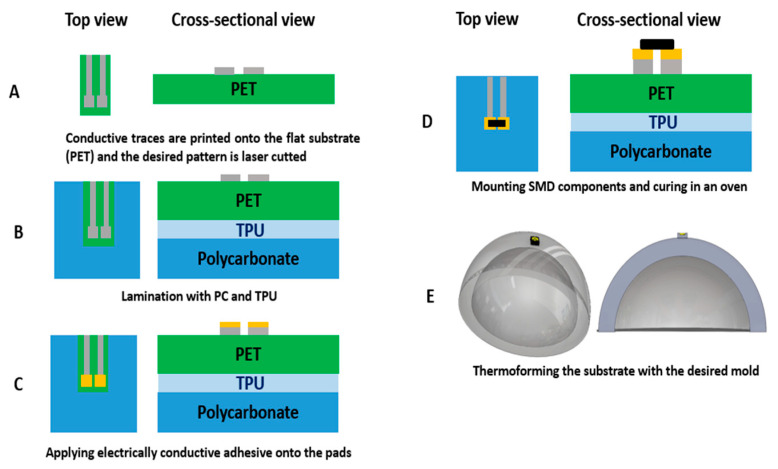
The fabrication process for adding the non-stretchable structure in the in-mold electronic fabrication process. (**A**) Conductive traces are printed onto the flat PET substrate using ME614 (Chimie Tech Services, Antony, France) ink, (**B**) The desired pattern for the non-stretchable structure is shaped by using laser ablation. (**C**) The fabricated substrate is laminated with a specific temperature and pressure profile with polycarbonate and TPU as the adhesive layer (Figure 12D). (**D**) electronic components are mounted on the substrate using a conductive adhesive. (**E**) The substrate is being thermoformed using a specific mold.

**Figure 12 micromachines-14-02248-f012:**
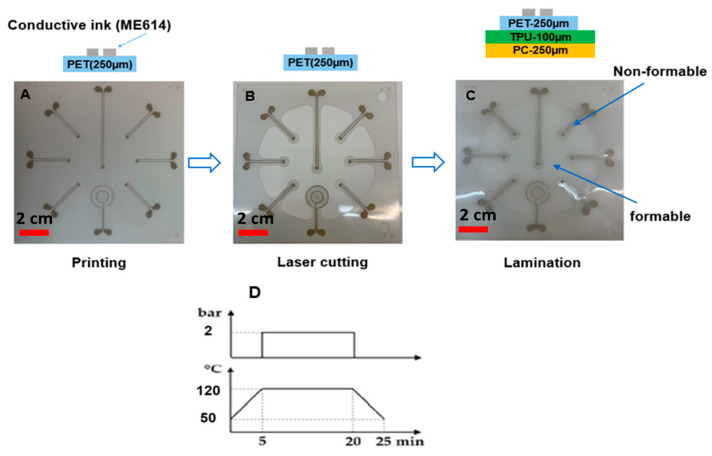
(**A**) Printing the conductive track on PET by using screen printing. (**B**) Laser cutting the printed sample to make the non-stretchable parts. (**C**) Lamination of the PET substrate with polycarbonate and TPU to make a thermoformable stack. (**D**) The lamination profile.

**Figure 13 micromachines-14-02248-f013:**
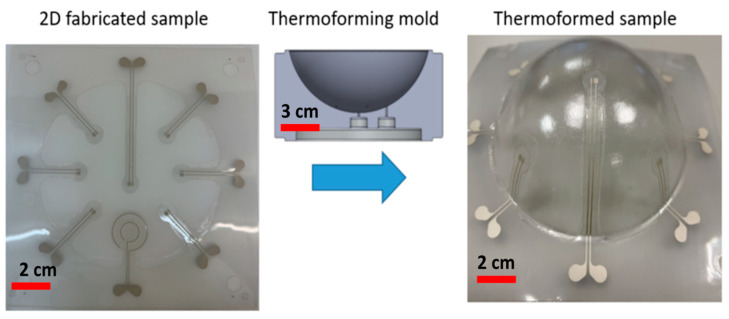
Thermoforming of the sample.

**Figure 14 micromachines-14-02248-f014:**
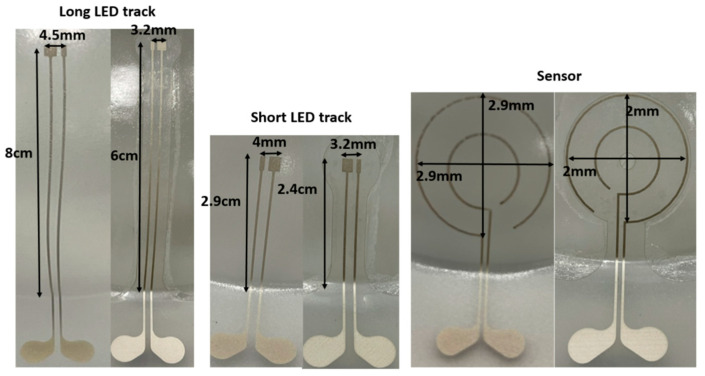
Measurement results of the thermoformed sample using a 60% elongation mold. Comparison between the typical fabrication process (the typical fabrication process is actually the “No mask” production process of the heat-mask methodology) and our proposed approach to fabricating in-mold electronics.

**Figure 15 micromachines-14-02248-f015:**
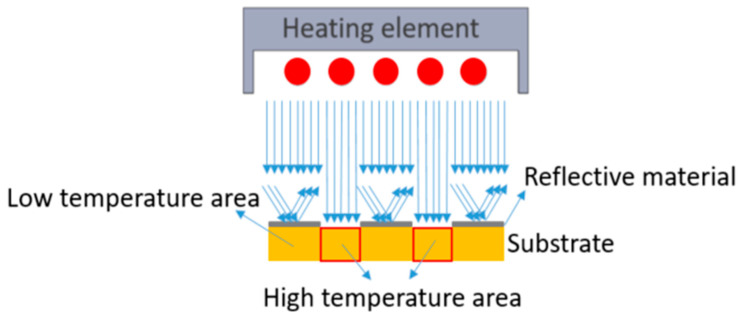
A substrate with a printed-reflective layer and temperate difference in the substrate.

**Figure 16 micromachines-14-02248-f016:**
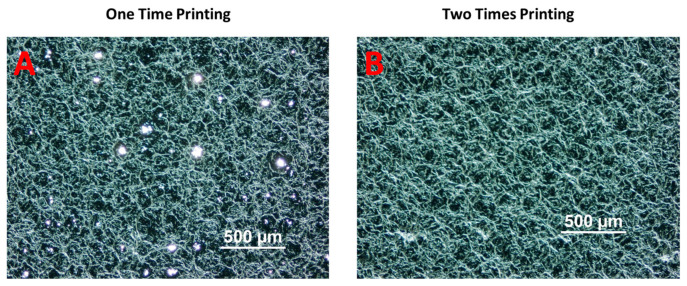
(**A**) The result of printed layer after one-time printing. (**B**) The result of the printed layer after two-times printing.

**Figure 17 micromachines-14-02248-f017:**
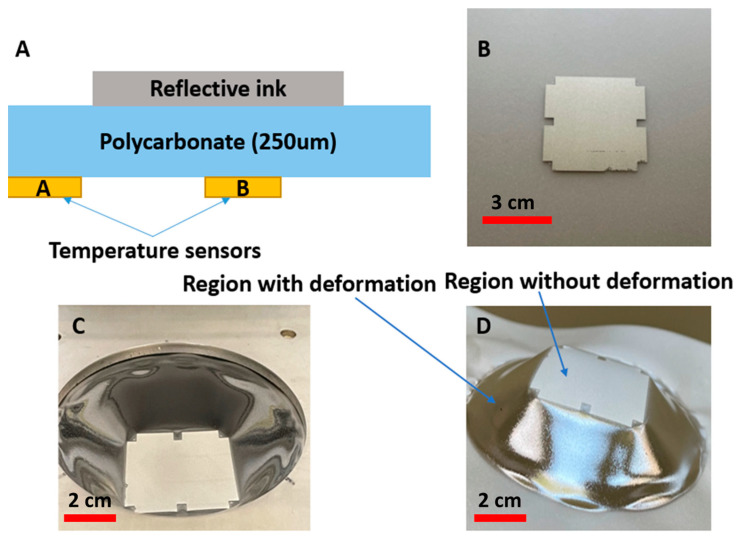
(**A**) Cross-sectional view of the printed stack. (**B**) The reflective material printed on the substrate. (**C**) The sample after being irradiated in the thermoforming machine. (**D**) The sample after being irradiated and removed from the thermoforming machine, in which it is apparent that in the region where a reflective layer is printed, there is no deformation as a result of the low temperature (the deformation in (**C**,**D**) is caused only by gravity, not by applying a vacuum and a forming mold).

**Figure 18 micromachines-14-02248-f018:**
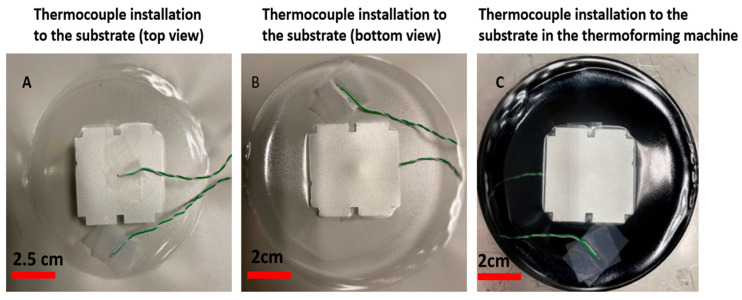
(**A**) Thermocouple installation to the substrate (top view). (**B**) Thermocouple installation to the substrate (bottom view). (**C**) Thermocouple installation to the substrate in the thermoforming machine.

**Figure 19 micromachines-14-02248-f019:**
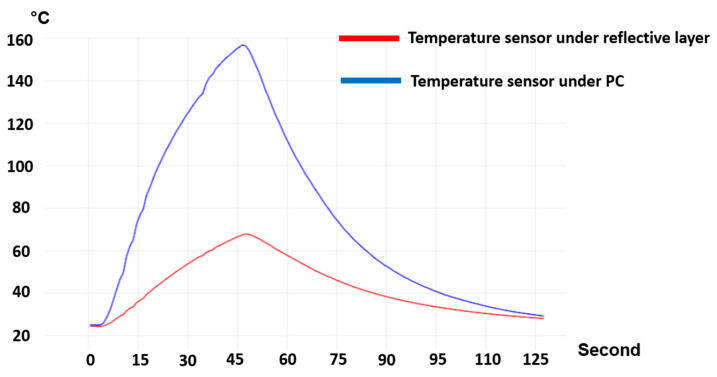
The temperature profile of the area where there is a reflective layer in comparison with the area where there is no reflective layer.

**Figure 20 micromachines-14-02248-f020:**
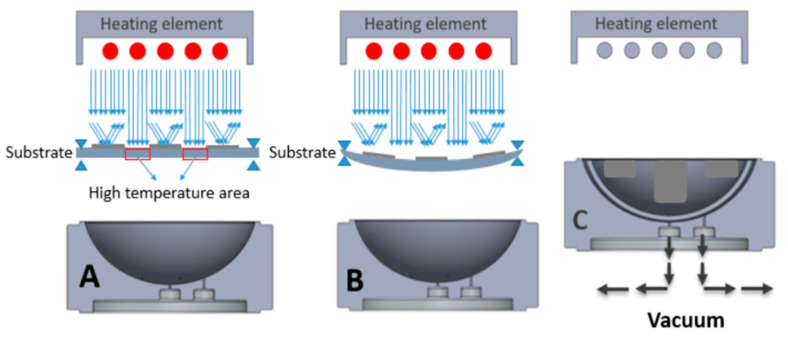
The effects of printing a reflective layer on the substrate on light absorption in the thermoforming process. (**A**) Heating process, (**B**) Starting the vacuum, (**C**) Thermoforming.

**Figure 21 micromachines-14-02248-f021:**
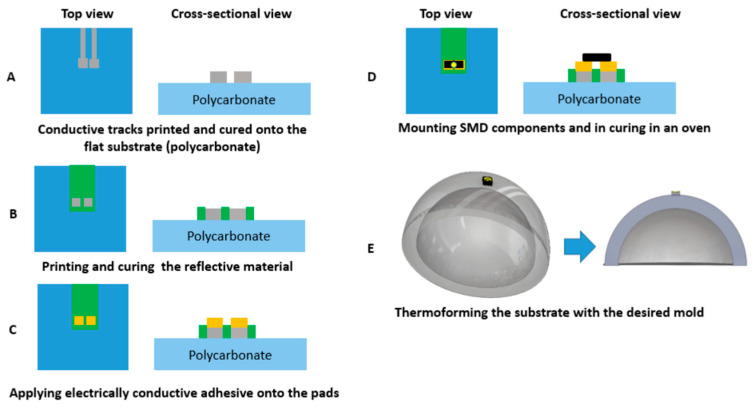
Fabrication process: (**A**) Printing conductive traces onto the flat substrate using ME614 and curing in the air convection oven at 90 °C for 20 min. (**B**) Printing the reflective layer (non-conductive mirror (Pröll GmbH) and curing in the oven 90 °C for 20 min. (**C**) Applying electrically conductive adhesive (LOCTITE CE 3103WLV)) onto the pads. (**D**) Mounting SMD components and curing in the oven at 120 °C for 30 min. (**E**) Thermoforming the substrate with the desired mold.

**Figure 22 micromachines-14-02248-f022:**
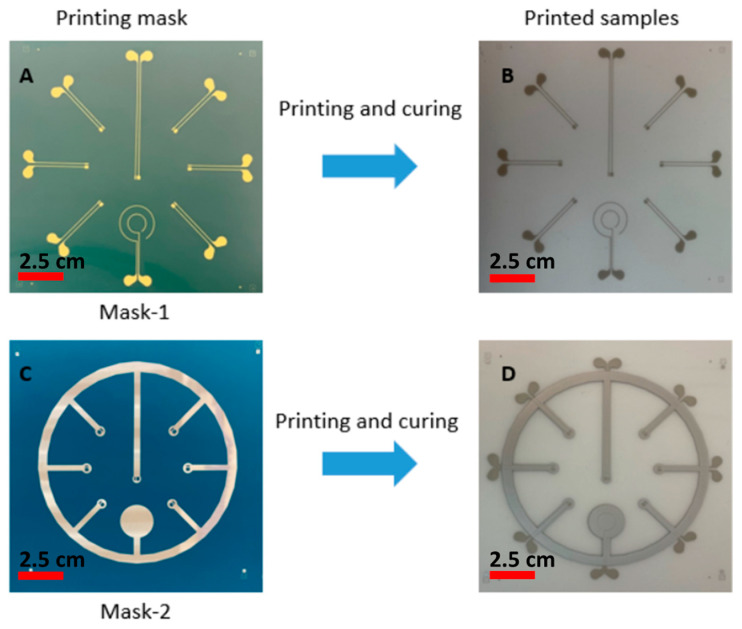
(**A**) Screen-printing mask (Mask-1). (**B**) Printing result on polycarbonate using ME164 as the conductor ink by using Mask-1. (**C**) Screen-printing mask (Mask-2). (**D**) Printing result on substrate (**B**) of non-conductive mirror (Pröll GmbH) as the reflective material, using Mask-2.

**Figure 23 micromachines-14-02248-f023:**
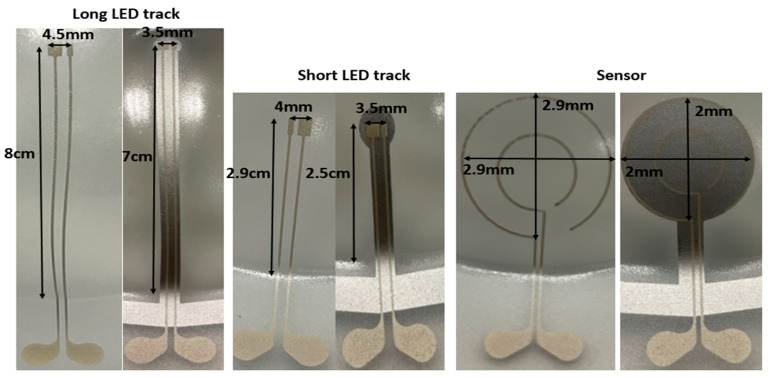
Measurement results of the thermoformed samples (for 3 different designs) using a 60% elongation mold. Comparison between the thermoformed samples with (right-hand picture for each design) and without (left-hand picture for each design) a reflective layer. Measured lengths are listed in Table 3.

**Figure 24 micromachines-14-02248-f024:**
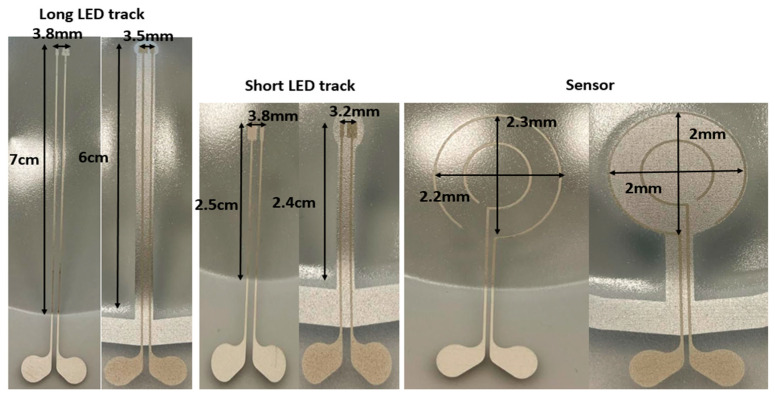
Measurement results of the thermoformed sample using a 16% elongation mold. Comparison between the thermoformed samples with and without a reflective layer.

**Table 1 micromachines-14-02248-t001:** Typical physical properties of ink (DuPont ME614).

Property	Value
Resistivity (mΩ/sq/mil)	≤40
Maximum stretch capacity (1 mm wide trace) (%)	~60
Coverage (c$m^{2}/g$) [using screen type 325 mesh polyester]	200

**Table 2 micromachines-14-02248-t002:** Printing parameters.

Parameters	Value
Printing speed forward (mm/s)	20
Printing speed backward (mm/s)	50
Printing press forward/backward (N)	100
Separation way (mm)	2
Separation speed (mm/s)	0.2

**Table 4 micromachines-14-02248-t004:** Measurement result on the thermoformed sample with a 16% elongation mold.

	No-Mask	Metal Mask	2D Length before Thermoforming
Long LED track	8 cm	6 cm	6 cm
Short LED track	2.7 cm	2.4 cm	2.4 cm
Sensor diameter	2.9 cm	2 cm	2 cm
Long track pad	4.5 mm	3.2 mm	3.2 mm
Short track pad	4 mm	3.2 mm	3.2 mm

**Table 5 micromachines-14-02248-t005:** Resistance measurement of the printed tracks before and after thermoforming by a (semi-sphere) mold (60% and 16% elongation).

	$R_{1}(Ω)$	$R_{2}(Ω)$	$R_{3}(Ω)$	$R_{4}(Ω)$	$R_{5}(Ω)$	$R_{6}(Ω)$	$R_{Average}(Ω)$
$R_{2D}$	5.36	6.32	5.43	6.39	4.88	5.66	5.67
$R_{3D}$ (16%) with Mask	6.2	6.1	5.8	5.5	6.1	6.01	5.95
$R_{3D}$ (60%) with Mask	6.56	7.46	5.21	6.48	6.91	5.58	6.36
$R_{3D}$ (16%) without Mask	7.53	11	8.17	7.92	9.09	8.16	8.64
$R_{3D}$ (60%) without Mask	ꝏ	ꝏ	ꝏ	ꝏ	ꝏ	ꝏ	ꝏ (open)

**Table 6 micromachines-14-02248-t006:** Measurement result on the thermoformed sample with a 60% elongation mold.

	Typical Fabrication Method	Our Proposed Method	2D Length before Thermoforming
Long LED track	8 cm	6 cm	6 cm
Short LED track	2.9 cm	2.4 cm	2.4 cm
Sensor diameter	2.9 cm	2 cm	2 cm
Long track pad	4.5 mm	3.2 mm	3.2 mm
Short track pad	4 mm	3.2 mm	3.2 mm

**Table 7 micromachines-14-02248-t007:** Resistance measurement of the printed tracks before and after thermoforming by a (semi-sphere) mold (60% elongation).

	$R_{1}(Ω)$	$R_{2}(Ω)$	$R_{3}(Ω)$	$R_{4}(Ω)$	$R_{5}(Ω)$	$R_{6}(Ω)$	$R_{Average}(Ω)$
$R_{2D}$	3.01	2.96	2.75	3.76	3.6	2.89	3.16
$R_{3D}$ (60%) with PET	3.1	3.01	2.86	3.6	3.52	3.42	3.25
$R_{3D}$ (60%) without PET	ꝏ	ꝏ	ꝏ	ꝏ	ꝏ	ꝏ	ꝏ (open)

**Table 8 micromachines-14-02248-t008:** Measurement results on the thermoformed sample using a 60% elongation mold.

	No-Reflective Layer	Reflective Layer	2D Length before Thermoforming
Long LED track	8 cm	7 cm	6 cm
Short LED track	2.9 cm	2.5 cm	2.4 cm
Sensor diameter	2.9 cm	2 cm	2 cm
Long track pad	4.5 mm	3.5 mm	3.2 mm
Short track pad	4 mm	3.5 mm	3.2 mm

**Table 9 micromachines-14-02248-t009:** Measurement results on the thermoformed sample using a 16% elongation mold.

	No-Reflective Layer	Reflective Layer	2D Length before Thermoforming
Long LED track	7 cm	6 cm	6 cm
Short LED track	2.5 cm	2.4 cm	2.4 cm
Sensor diameter	2.2 cm	2 cm	2 cm
Long track pad	3.8 mm	3.2 mm	3.2 mm
Short track pad	3.8 mm	3.2 mm	3.2 mm

**Table 10 micromachines-14-02248-t010:** Resistance measurements of the printed tracks before and after thermoforming by a (semi-sphere) mold (60% and 16% elongation mold).

	$R_{1}(Ω)$	$R_{2}(Ω)$	$R_{3}(Ω)$	$R_{4}(Ω)$	$R_{5}(Ω)$	$R_{6}(Ω)$	$R_{Average}(Ω)$	STD
$R_{2D}$	3.82	4.12	4.01	4.18	4.26	4.20	4.09	0.14
$R_{3D}$ (16%) with reflective layer	3.79	4.71	4.05	4.20	4.67	4.28	4.28	0.32
$R_{3D}$ (60%) with reflective layer	4.72	4.74	4.66	4.80	5.01	5.24	4.86	0.20
$R_{3D}$ (16%) without reflective layer	7.53	11	8.17	7.92	9.09	8.16	8.64	1.15
$R_{3D}$ (60%) without reflective layer	ꝏ	ꝏ	ꝏ	ꝏ	ꝏ	ꝏ	ꝏ (open)	0

**Table 11 micromachines-14-02248-t011:** Comparison of the three approaches.

	Heat Mask	Printed Mask	Structured Foil
Positioning accuracy (%)	Error < 1% (high)	Error < 16% (low)	Error < 1% (high)
Resistance changes (%)	12%	18%	18%
2D design complexity	Low	Low	Low
Extra fabrication steps	No	Yes	Yes
Spatial frequency exposed/non-exposed areas	Low (2 cm or lower)	High (5 cm or higher)	High (5 cm or higher)

## Data Availability

Data are contained within the article.

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
