# Peer review of "Methods to Improve Accuracy of Electronic Component Positioning in Thermoformed Electronics"

_micromachines, 2023, doi:10.3390/mi14122248_

Round 1

Reviewer 1 Report

Comments and Suggestions for Authors

This paper reports the methods for thermoformed electronics to improve accuracy of electric componet. The author propesed three methods to prevent temperature rise of the substrate during thermoforming process. It is of interestin to the readers on printed flexible electronics. However, there are some missing conditions in the experiment conditions. We cannot recommend articles to be published in Micromachines in the current form until the following points are cleared.

1) It is better to add a scale bar to the photos and figures in Figures 3, 4, 6, 7, 12, 13, 16, 17, 21.

2) In line 121, it is desirable to describe the value of glass transition temerature of PC.

3) There are a lot of "Error! Reference source not found" in the text.

4) In line 192, there is "a 15% elongation mold" which is shown in figures captions of 9 and 23, and tables 5 and 10.

5) In line 198, A2 and A1 is represented as Area2 and Area1 in the above equation.

6) In line 331, it is preferable to describe the thickness of reflective layer or the size of mesh using screen printing for reproducibility.

Reviewer 2 Report

Comments and Suggestions for Authors

A heat mask, a higher Tg thermoplastic substrate, and a reflective material on top of the printed tracks are adopted for accurate electronic component positioning for thermoformed electronics. The following comments need to be addressed before the next consideration: 1) In terms of the thermoformed electronics community, what are the primary goals in precision, the key process data of special device packages, and the pros and cons difference between the proposed three novel approaches of this work? 2) The introduction needs a few more recent studies, especially since 2020. 3) In line 146, it should be Fig4 instead of Fig5. The reason for the difference between Fig 4a and Fig 4b needs to be clarified. The most valuable and interesting point is the 3D metal line space and position controlled with a 2D mask, but it is ignored. 4) In line 195, the proposed equation of the Elongation is an approximate calculation; how about the accuracy? 5) Besides the material’s thermal properties, the relationship between Tg and the oven temperature should be clearly demonstrated. 6) More details of the resistance test method and device should be added. 7) In the main text, ALL number of references is missing, and errors are reported.

Comments on the Quality of English Language

Can be further inproved. 

Round 2

Reviewer 2 Report

Comments and Suggestions for Authors

After revison, the paper can be accepted for publication.